# Testing persuasive messages about booster doses of COVID-19 vaccines on intention to vaccinate in Australian adults: A randomised controlled trial

**Maryke S. Steffens**[1,2]*, **Bianca Bullivant**[1,2], **Jessica Kaufman**[3,4], **Catherine King**[1,2], **Margie Danchin**[3,4,5], **Monsurul Hoq**[3,6,7], **Mathew D. Marques**[8]

1 National Centre for Immunisation Research and Surveillance, Kids Research, Sydney Children's Hospitals Network, Westmead, NSW, Australia, 2 Faculty of Medicine and Health, The Children's Hospital at Westmead Clinical School, The University of Sydney, Sydney, NSW, Australia, 3 Vaccine Uptake Research Group, Murdoch Children's Research Institute, Melbourne, VIC, Australia, 4 Department of Paediatrics, University of Melbourne, Melbourne, VIC, Australia, 5 Department of General Medicine, Royal Children's Hospital, Melbourne, VIC, Australia, 6 Clinical Epidemiology and Biostatistics Unit, Murdoch Children's Research Institute, Parkville, Australia, 7 The National Child Health Poll, The Royal Children's Hospital, Parkville, Australia, 8 School of Psychology and Public Health, La Trobe University, Melbourne, VIC, Australia

* maryke.steffens@health.nsw.gov.au

## Abstract

### Introduction

Achieving high COVID-19 vaccine booster coverage is an ongoing global challenge. Health authorities need evidence about effective communication interventions to improve acceptance and uptake. This study aimed to test effects of persuasive messages about COVID-19 vaccine booster doses on intention to vaccinate amongst eligible adults in Australia.

### Methods

In this online randomised controlled trial, adult participants received one of four intervention messages or a control message. The control message provided information about booster dose eligibility. Intervention messages added to the control message, each using a different persuasive strategy, including: emphasising personal health benefits of booster doses, community health benefits, non-health benefits, and personal agency in choosing vaccination. After the intervention, participants answered items about COVID-19 booster vaccine intention and beliefs. Intervention groups were compared to the control using tests of two proportions; differences of ≥5 percentage points were deemed clinically significant. A sub-group analysis was conducted among hesitant participants.

### Results

Of the 487 consenting and randomised participants, 442 (90.8%) completed the experiment and were included in the analysis. Participants viewing messages emphasising non-health benefits had the highest intention compared to those who viewed the control message

**Data Availability Statement:** The data files are available from the Open Science Framework at osf.io/a6e5s.

**Funding:** MS, BB, and CK received funds from NSW Health (https://www.health.nsw.gov.au/) to conduct this research. The funders had no role in study design, data collection and analysis, decision to publish, or preparation of the manuscript.

**Competing interests:** The authors have declared that no competing interests exist.

(percentage point diff: 9.0, 95% CI -0.8, 18.8, $p = 0.071$). Intention was even higher among hesitant individuals in this intervention group compared to the control group (percentage point diff: 15.6, 95% CI -6.0, 37.3, $p = 0.150$). Conversely, intention was lower among hesitant individuals who viewed messages emphasising personal agency compared to the control group (percentage point diff: -10.8, 95% CI -33.0, 11.4, $p = 0.330$), although evidence in support of these findings is weak.

## Conclusion

Health authorities should highlight non-health benefits to encourage COVID-19 vaccine booster uptake but use messages emphasising personal agency with caution. These findings can inform communication message development and strategies to improve COVID-19 vaccine booster uptake.

**Clinical trial registration:** Registered with the Australian New Zealand Clinical Trials Registry (ACTRN12622001404718); trial webpage: https://www.anzctr.org.au/ACTRN12622001404718.aspx

## Introduction

COVID-19 vaccination has been critical for controlling the COVID-19 pandemic by protecting vulnerable individuals from severe disease, safeguarding health systems and helping return society to normal functioning [1, 2]. Booster doses of COVID-19 vaccines (additional doses given after completion of the two-dose primary course) are necessary to provide ongoing immunity to SARS-CoV-2 and offer increased protection against severe disease [3].

Suboptimal uptake of COVID-19 booster doses has occurred globally. In Australia, completion of a primary course of COVID-19 vaccine for individuals ≥16 years is >96%, but booster dose (third dose) coverage has stalled at just over 70% [4]. In other countries, the United Kingdom (UK) has achieved approximately 70% booster dose coverage in the eligible population [5], while Europe and the United States (US) have achieved just over 50% and 43% respectively [6, 7].

While many factors can contribute to low vaccine uptake, including both access and acceptance barriers, low motivation may continue to act as a barrier to uptake if not addressed [8]. This has been the experience with other pandemic vaccines, for example the 2009 H1N1 influenza pandemic vaccine [9, 10]. There are a range of factors that influence motivation to receive a COVID-19 vaccine. Both personal and collective health benefits have been found to shape acceptance and motivation [11–13], building on evidence from strategies to increase vaccine uptake more broadly [14–16]. Non-health benefits, i.e. the broader benefits of vaccination beyond direct protection from disease such as ability to travel, have been found to be motivating factors [12, 17]. Attaching importance to certain moral values, such as individual liberties and the agency or freedom to make personal health decisions, have been linked to hesitancy about COVID-19 vaccines [18] as well as routine vaccines [19].

Considering these findings, emphasising motivational factors in persuasive messaging could potentially be a successful strategy to support higher intention to receive a COVID-19 vaccine. There is evidence that persuasive messages can support higher intentions towards vaccination in general [8, 20, 21]. For COVID-19 vaccination specifically, a systematic review has found that messages emphasising personal or community health benefits of vaccination can be

effective in supporting higher intentions, although results are mixed [22]. Three survey experiments in US and UK adults comparing the effect of a range of messages on intention to receive a COVID-19 vaccine found that messages emphasising personal health benefits of vaccination have a larger effect than those emphasising benefits to others [13, 23, 24]. By contrast, another large survey experiment in US adults found that messages emphasising non-health benefits (such as freedom from public health restrictions) and community benefits were effective. The community messages were the most effective of all, while those emphasising personal health benefits were not more effective than the control [25]. Evidence of the effectiveness of messaging promoting altruistic behaviour (i.e. behaviour beneficial to the community) on COVID-19 vaccine intention has also been found in other experiments in US and UK adults [26, 27]. There is limited research investigating the effectiveness of messages emphasising personal agency in making COVID-19 vaccination decisions.

Given the current state of evidence, it remains unclear what types of messages have the greatest effect on intention to receive a booster dose of COVID-19 vaccine. This is especially the case in populations that have experienced relatively low risk of encountering COVID-19, combined with restrictive public health measures. These were the conditions experienced in Australia at the time of this experiment, where COVID-19 case numbers were relatively low during the first two years of the pandemic, while public health restrictions, such as border closures and lockdowns, were applied liberally due to the country's pursuit of a COVID-19 elimination strategy [28]. Such evidence can inform communications from health authorities and other stakeholders to encourage uptake of COVID-19 booster doses, especially written communications (e.g. emails), which were used to communicate with the public in Australia during the COVID-19 vaccine rollout. This study aimed to compare the effect of persuasive messages on intention to get a booster dose of COVID-19 vaccine in Australian adults. This study is part of a larger study investigating factors influencing COVID-19 vaccine acceptance in Australia and elsewhere [11, 12, 29, 30] and developing messaging to support acceptance of COVID-19 vaccines in various populations [31].

## Methods

### Experimental design

This was a parallel group, randomised, controlled, online experiment comparing intention to receive a COVID-19 vaccine booster dose and beliefs about COVID-19 vaccine booster doses between subjects after receiving persuasive message interventions. This study obtained ethics approval from the Sydney Children's Hospitals Network Human Research Ethics Committee (2021/ETH00181). The study aims, methods, and data analysis plan were pre-registered on the Open Science Framework (https://osf.io/xajmz). In response to the skewed distribution of responses, a post hoc variation to the pre-registration was made to analyse intention to vaccinate and participant responses to belief items as dichotomous variables. See S1 File for original frequency distribution of intention to vaccinate. The trial was retrospectively registered with the Australian New Zealand Clinical Trials Registry (ACTRN12622001404718, https://www.anzctr.org.au/ACTRN12622001404718.aspx).

### Participants

Participants were adults 18 years or older residing in Australia who had received at least one primary dose of a COVID-19 vaccine but had not yet received a booster (third) dose and had access to the internet. Research company Quality Online Research (QOR) recruited a random sample of participants via email invitation from its accredited online panel. The QOR panel has >85,000 active members; original panel members were recruited via the Australia Post

Lifestyle Survey, distributed to all Australian households. Ongoing recruitment is by invitation only. The panel reflects Australian Bureau of Statistics census data by age, gender, and state. Participants took part in the study via an online portal hosted by the research company and were offered points as incentive for participation equivalent to between AUD$1.00-$2.00, redeemable as payouts by PayPal, eGift cards or cheques. Participants gave written (digital) informed consent. Participants were recruited between 17–24 December 2021, with recruitment stopping when targets were met.

For context, at the time of data collection, the Australian population had experienced multiple rounds of restrictions on movement, limits on indoor and outdoor gatherings, closure of restaurants, gyms and non-essential retail, and interstate and international border closures. Approximately two months prior to data collection (in October 2021), people living in the states of New South Wales and Victoria had exited strict and lengthy lockdowns in response to the Delta wave. In December 2021, domestic border closures had started to lift, however strict international border closures, put in place in March 2020 to prevent people from leaving and entering the country, were still in place at the time. In December 2021, Australia was anticipating the occurrence of a further Omicron wave, with cases rising rapidly [32, 33].

## Pre-intervention survey items

After consenting, participants provided demographic information (age, gender, education, and state of residence). They were asked to respond to screening items about their COVID-19 vaccination status ('Have you received a first/second/third dose of a COVID-19 vaccine?', response options = yes/no). Participants were asked to indicate their hesitancy towards COVID-19 vaccines with a single item ('How much do you agree with the following statement: "I feel hesitant about COVID-19 vaccines"', response options were on a 5-point scale from strongly disagree to strongly agree) informed by previous vaccination research [34, 35]. This item was used to categorise the participants into vaccine hesitant and accepting participants. Responses were recoded where (strongly agree, slightly agree, neither agree nor disagree) = hesitant, and (slightly disagree, strongly disagree) = accepting.

## Intervention

Participants received one of four intervention messages according to their randomised group. The intervention was a short piece of written material (a message of approximately 70–140 words) designed to encourage uptake of booster doses of COVID-19 vaccines. The material was modified from a public email communication campaign disseminated by an Australian health authority in November 2021. These modifications were made by the research team, based on the current state of evidence on the types of messages that may have an effect on intention to vaccinate. The control message informed recipients of eligibility requirements for a booster dose of COVID-19 vaccine. The four intervention messages added to this message, each using a different persuasive strategy. Table 1 shows the full intervention messages. The *'Personal health benefits'* message emphasised the reduction in risk of becoming infected, sick or dying that getting a booster dose of COVID-19 vaccine offers. The *'Community health benefits'* message emphasised the altruistic nature of receiving a COVID-19 vaccine, i.e., the reduction in risk of giving the virus to family members or people in the community. The *'Non-health benefits'* message emphasised the broader benefits that getting vaccinated could bring such as the possibility of freedom from future public health restrictions. The *'Personal agency'* message emphasised the control over one's health gained by choosing vaccination. Survey software required participants to view the intervention text for a minimum of 30 seconds.

**Table 1. Intervention messages.**

| Message name | Full text |
|---|---|
| *Control* | **Get your COVID-19 vaccine booster shot**<br>If your second dose of a COVID-19 vaccine was more than 5 months ago, you can now receive a booster vaccination.<br>You can check when you received your second dose by looking at your immunisation history statement or COVID-19 vaccination certificate. |
| *Personal health benefits* | **Protect your health, get your COVID-19 vaccine booster shot**<br>If your second dose of a COVID-19 vaccine was more than 5 months ago, you can now receive a booster vaccination.<br>Your protection from COVID-19 after vaccination reduces over time. Getting a booster vaccination will reduce the risk that you get infected, become very sick, or die from COVID-19.<br>Booster doses of COVID-19 vaccine give you protection, no matter how old you are.<br>You can check when you received your second dose by looking at your immunisation history statement or COVID-19 vaccination certificate.<br>Remember, getting a booster dose of COVID-19 vaccine is the best way to keep protecting yourself. |
| *Community health benefits* | **Protect people you care about, get your COVID-19 vaccine booster shot**<br>If your second dose of a COVID-19 vaccine was more than 5 months ago, you can now receive a booster vaccination.<br>Protection from COVID-19 after vaccination reduces over time. No matter how old you are, getting a booster vaccination not only protects you, it also reduces your risk of giving the virus to your family members or people in the community, who could get sick and die from COVID-19.<br>You can check when you received your second dose by looking at your immunisation history statement or COVID-19 vaccination certificate.<br>Remember, getting a booster dose of COVID-19 vaccine is the best way to keep protecting all of us. |
| *Non-health benefits* | **Help get life back to normal. Get your COVID-19 vaccine booster shot**<br>If your second dose of a COVID-19 vaccine was more than 5 months ago, you can now receive a booster vaccination.<br>COVID-19 has stopped us from living our lives as freely as we used to. We've been locked down, and not been able to travel, go to weddings or funerals, and see family and friends. When you get a booster dose of COVID-19 vaccine, you're helping to reduce the chance that restrictions return.<br>You can check when you received your second dose by looking at your immunisation history statement or COVID-19 vaccination certificate.<br>Remember, while you can't do it alone, getting a booster vaccination is the best way for you to help make sure we can all keep living freely. |
| *Personal agency* | **Take control of your health. Get your COVID-19 vaccine booster shot**<br>If your second dose of a COVID-19 vaccine was more than 5 months ago, you can now receive a booster vaccination.<br>Getting a booster vaccination is not mandatory. It's a personal choice–one that gives you control of your health and lets you protect the people you care about.<br>We all want the freedom to make our own decisions about our health. When you get a booster dose of COVID-19 vaccine, you're taking charge and choosing the best option to protect yourself, your family, and your community.<br>You can check when you received your second dose by looking at your immunisation history statement or COVID-19 vaccination certificate.<br>Remember, it's in your hands to make the right decision for yourself and the people you care about. |

## Post-intervention outcome measures

Immediately after the intervention, participants responded to items measuring outcomes. The primary outcome measure was intention to receive a COVID-19 vaccine booster dose. This was assessed with a single item ('How likely is it that you will get a booster dose of COVID-19 vaccine?', with 5 response options: definitely, probably, I'm not sure, probably not, definitely not), consistent with survey questions used in previous research [36]. Responses were

transformed into a dichotomous variable where (definitely, probably) = 'intends to receive a COVID-19 vaccine' and (I'm not sure, probably not, definitely not) = 'does not intend to receive a COVID-19 vaccine'. Participants then answered an attention check question ('Please select strongly disagree for this item', response options ranged from strongly disagree to strongly agree).

Secondary outcome measures were beliefs about COVID-19 vaccine booster doses. These were assessed by asking participants to indicate their agreement with belief statements about COVID-19 vaccine booster dose safety, effectiveness, necessity for protecting one's own health, necessity for protecting others' health, and risks associated with not vaccinating (response options were a 5-point scale: strongly disagree, slightly disagree, neither agree nor disagree, slightly agree, strongly agree. See S1 File for belief statements). Responses were transformed into dichotomous variables where (Strongly disagree, Slightly disagree, Neither agree nor disagree) = 'does not agree with belief' and (Strongly agree, Slightly agree) = 'agrees with belief'.

## Sample size

The study aimed to recruit 480 participants to ensure a sample size of 430 participants, allowing for a drop-out/poor quality response rate of approximately 10%. The sample size was calculated using G*Power (an a priori power analysis tool) to estimate an effect size of Cohen's d = 0.2 (the difference between two independent means) [37] with a specified power of 80% at 0.01 level of significance using analysis of variance (ANOVA) tests to compare outcomes between intervention groups, as specified in the pre-registered protocol. In line with the post hoc variation to analyse participant's intention to vaccinate and belief items as dichotomous variables, the sample size of 85 participants per intervention group was adequate to estimate a difference in proportion of 15 percentage points or more between two groups using two-sample test of proportion (z-test) with a level of significance of 0.05 and power of 80%. Participants with incomplete surveys or who failed to answer the quality control attention check question correctly were excluded.

## Randomisation

Participants were randomly assigned at recruitment via a randomisation sequence embedded within the online system to receive one of the four intervention messages or the control message. Because some participants were excluded after randomisation (they either failed the attention check or did not complete the experiment, see Fig 1), a second randomisation sequence was generated to ensure intervention and control groups were of equal size. This sequence prioritised the least filled group when 80% of the recruitment was complete.

## Statistical methods

The difference in proportion of participants intending to receive a COVID-19 vaccine and agreeing with COVID-19 vaccine booster dose belief statements was compared between the control group and each of the four intervention groups using tests of two proportions (z test). A subgroup analysis included data from vaccine hesitant participants only.

Previous studies have shown that an effect size of ≥5 percentage points may have practical (clinical) significance [38–40]. Hence, when reporting differences in intention and beliefs for each intervention group compared to the control group, differences of ≥5 percentage points were considered practically meaningful from a public health perspective, regardless of the p value [41]. The difference in proportion was interpreted with respect to the upper and lower limit of the confidence intervals.

All analyses were conducted using SPSS (Statistical Package for Social Sciences, version 27).

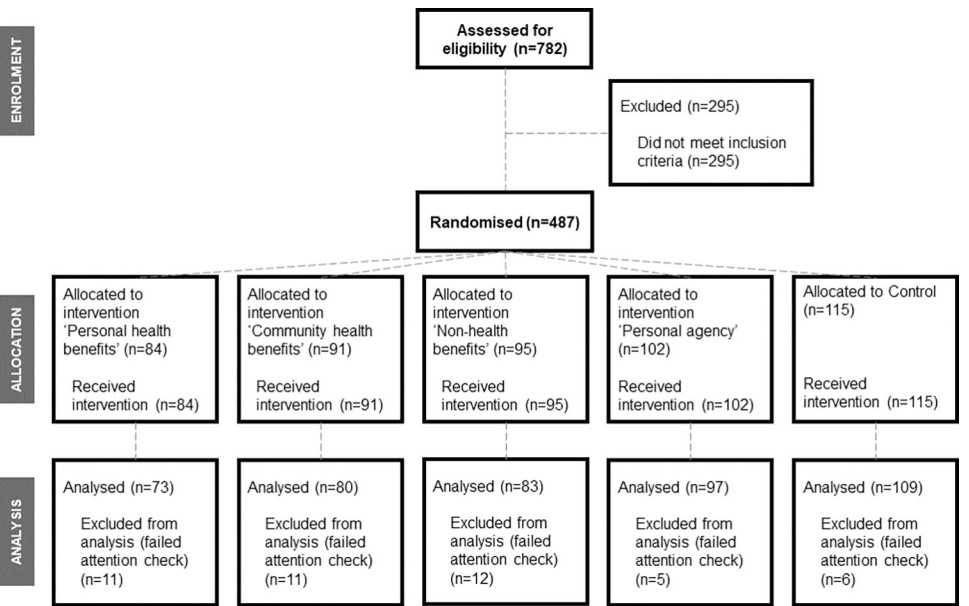

**Fig 1. Flow diagram showing progress of participants through the online experiment.**

### Role of the funding source

This research was funded by a grant from NSW Health. The sponsor had no role in the study design; in the collection, analysis and interpretation of data; in the writing of the report; or in the decision to submit the article for publication.

## Results

Of the 487 participants who consented and were randomised, 442 (90.8%) completed the experiment, answered the attention check question correctly, and were included in the analysis. Fig 1 shows the progress of participants through the online experiment.

Participants had a mean age of 51.14 years (*SD* 16.32); 51.1% (226/442) were female; 97.5% (431/442) had reported having received a second dose of COVID-19 vaccine; 37.8% (167/442) reported feeling hesitant about COVID-19 vaccines. Table 2 shows participant characteristics between groups, which are similar in age, gender, education, state, completion of second dose, or hesitancy between groups.

### Primary outcome (Intention to vaccinate)

Most participants (84.6%, 374/442) indicated a positive intention to get a booster dose of COVID-19 vaccine. Table 3 shows intention to vaccinate, compared between groups. All intervention groups had a qualitatively higher intention to vaccinate compared to the control group except one (see Fig 2). Participants who viewed the message emphasising non-health benefits showed the largest difference compared to the control group (percentage point diff: 9.0, 95% CI -0.8, 18.8, *p* = 0.071), although the evidence in support of this finding is weak. Participants who viewed the message emphasising personal agency had a slightly lower intention compared to the control group (percentage point diff: -4.2, 95% CI -15.1, 6.7, *p* = 0.445), although there is a lack of evidence in support of this finding.

In the sub-analysis of hesitant participants, more than half of participants (64.7%; 108/167) intended to get a booster dose of COVID-19 vaccine; this proportion is lower than for the

**Table 2. Participant characteristics at baseline.**

| Group | All | Control | Personal health benefits | Community health benefits | Non-health benefits | Personal agency | |
|---|---|---|---|---|---|---|---|
| Total n | 442 | 109 | 73 | 80 | 83 | 97 | |
| **How old are you? n (%)** | | | | | | | |
| 18–24 | 21 (4.8) | 7 (6.4) | 4 (5.5) | 5 (6.3) | 2 (2.4) | 3 (3.1) | |
| 25–29 | 22 (5.0) | 7 (6.4) | 1 (1.4) | 5 (6.3) | 1 (1.2) | 8 (8.2) | |
| 30–39 | 81 (18.3) | 25 (22.9) | 16 (21.9) | 15 (18.8) | 15 (18.1) | 10 (10.3) | |
| 40–49 | 83 (18.8) | 20 (18.3) | 15 (20.5) | 13 (16.3) | 18 (21.7) | 17 (17.5) | |
| 50–59 | 86 (19.5) | 17 (15.6) | 17 (23.3) | 16 (20.0) | 15 (18.1) | 21 (21.6) | |
| 60–69 | 76 (17.2) | 17 (15.6) | 11 (15.1) | 10 (12.5) | 15 (18.1) | 23 (23.7) | |
| 70+ | 73 (16.5) | 16 (14.7) | 9 (12.3) | 16 (20.0) | 17 (20.5) | 15 (15.5) | |
| **What is your gender? n (%)** | | | | | | | |
| Female | 226 (51.1) | 53 (48.6) | 33 (45.2) | 40 (50.0) | 46 (55.4) | 54 (55.7) | |
| Male | 213 (48.2) | 55 (50.5) | 39 (53.4) | 40 (50.0) | 37 (44.6) | 42 (43.3) | |
| Not specified | 3 (0.7) | 1 (0.9) | 1 (1.4) | 0 (0.0) | 0 (0.0) | 1 (1.0) | |
| **What is the highest level of education or training you have completed? n (%)** | | | | | | | |
| Did not attend school | 2 (0.5) | 0 (0.0) | 0 (0.0) | 0 (0.0) | 1 (1.2) | 1 (1.0) | |
| Year 12 or below | 145 (32.8) | 33 (30.3) | 29 (39.7) | 26 (32.5) | 23 (27.7) | 34 (35.1) | |
| University degree | 175 (39.6) | 49 (45.0) | 23 (31.5) | 30 (37.5) | 35 (42.2) | 38 (39.2) | |
| Other non-school qualifications | 115 (26.0) | 25 (22.9) | 21 (28.8) | 24 (30.0) | 22 (26.5) | 23 (23.7) | |
| Prefer not to answer | 5 (1.1) | 2 (1.8) | 0 (0.0) | 0 (0.0) | 2 (2.4) | 1 (1.0) | |
| **Where do you currently live (Australian state or territory)? n (%)** | | | | | | | |
| NSW | 139 (31.4) | 37 (33.9) | 16 (21.9) | 21 (26.3) | 30 (36.1) | 35 (36.1) | |
| QLD | 82 (18.6) | 17 (15.6) | 16 (21.9) | 20 (25.0) | 10 (12.0) | 19 (19.6) | |
| VIC | 123 (27.8) | 31 (28.4) | 22 (30.1) | 26 (32.5) | 23 (27.7) | 21 (21.6) | |
| ACT | 2 (0.5) | 1 (0.9) | 0 (0.0) | 0 (0.0) | 0 (0.0) | 1 (1.0) | |
| SA | 41 (9.3) | 5 (4.6) | 9 (12.3) | 7 (8.8) | 7 (8.4) | 13 (13.4) | |
| WA | 39 (8.8) | 14 (12.8) | 8 (11.0) | 3 (3.8) | 9 (10.8) | 5 | (5.2) |
| TAS | 16 (3.6) | 4 (3.7) | 2 (2.7) | 3 (3.8) | 4 (4.8) | 3 (3.1) | |
| NT | 0 | 0 | 0 | 0 | 0 | 0 | |
| **Have you received a second dose of a COVID-19 vaccine? n (%)** | | | | | | | |
| Yes | 431 (97.5) | 105 (96.3) | 72 (98.6) | 77 (96.3) | 83 (100.0) | 94 (96.9) | |
| No | 11 (2.5) | 4 (3.7) | 1 (1.4) | 3 (3.8) | 0 (0.0) | 3 (3.1) | |
| **How much do you agree with the following statement: "I feel hesitant about COVID-19 vaccines." n (%)** | | | | | | | |
| Disagree | 275 (62.2) | 71 (65.1) | 49 (67.1) | 50 (62.5) | 50 (60.2) | 55 (56.7) | |
| Agree | 167 (37.8) | 38 (34.9) | 24 (32.9) | 30 (37.5) | 33 (39.8) | 42 (43.3) | |

**Table 3. Comparing intention to vaccinate between intervention groups and the control group.**

| Group | Total n | Definitely/probably intend on getting a booster dose | % | Diff[§] | 95% CI Lower | 95% CI Upper | p |
|---|---|---|---|---|---|---|---|
| Control | 109 | 90 | 82.6 | - | - | - | - |
| Personal health benefits | 73 | 63 | 86.3 | 3.7 | -7.2 | 14.7 | 0.500 |
| Community health benefits | 80 | 69 | 86.2 | 3.7 | -7.0 | 14.3 | 0.494 |
| Non-health benefits | 83 | 76 | 91.6 | 9.0* | -0.8 | 18.8 | 0.071 |
| Personal agency | 97 | 76 | 78.4 | -4.2 | -15.1 | 6.7 | 0.455 |
| All participants | 442 | 374 | 84.6 | | | | |

[§] Percentage point difference

*Indicates a difference of ≥5 percentage points between the intervention group and the control group

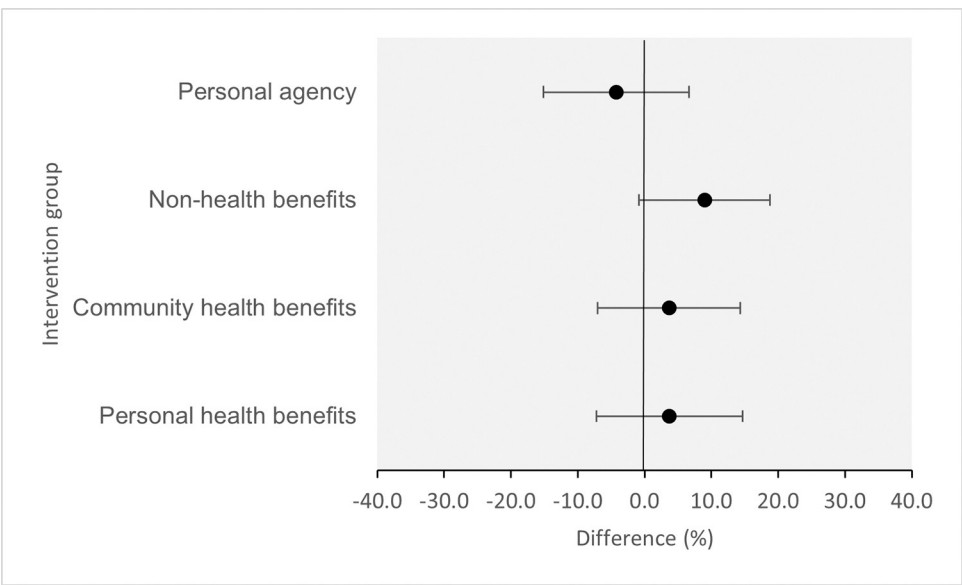

**Fig 2. Difference in intention to vaccinate in each intervention group compared to the control group.**

whole study population. Table 4 shows intention to vaccinate in hesitant participants, compared between groups. As with the whole population, all intervention groups had a qualitatively higher intention to vaccinate compared to the control group except one (see Fig 3). Participants who viewed the message emphasising non-health benefits showed a higher intention compared to the control group (percentage point diff: 15.6, 95% CI -6.0, 37.3, *p* = 0.150), while participants who viewed the personal agency message had a lower intention compared to the control group (percentage point diff: -10.8, 95% CI -33.0, 11.4, *p* = 0.330), although there is a lack of evidence in support of these findings. Results for non-hesitant participants are reported in S1 File.

## Secondary outcome (Beliefs)

Most participants agreed that booster doses are safe (71.7%, 317/442), prevent disease (75.3%, 333/442), protect their health (78.7%, 348/442), protect the health of others (77.4%, 342/442). A slightly smaller majority agreed that they were at risk of getting COVID-19 without a booster dose (63.8%, 282/442).

**Table 4. Comparing intention to vaccinate between intervention groups and the control group in hesitant participants (sub-analysis).**

| Group | Total n | Definitely/probably intend on getting a booster dose | % | Diff$^{§}$ | 95% CI Lower | 95% CI Upper | *p* |
|---|---|---|---|---|---|---|---|
| Control | 38 | 24 | 63.2 | - | - | - | - |
| Personal health benefits | 24 | 16 | 66.7 | 3.5 | -21.8 | 28.9 | 0.779 |
| Community health benefits | 30 | 20 | 66.7 | 3.5 | -21.8 | 28.9 | 0.764 |
| Non-health benefits | 33 | 26 | 78.8 | 15.6* | -6.0 | 37.3 | 0.150 |
| Personal agency | 42 | 22 | 52.4 | -10.8* | -33.0 | 11.4 | 0.330 |
| All participants | 167 | 108 | 64.7 | | | | |

$^{§}$ Percentage point difference

*Indicates a difference ≥5% between the intervention group and the control group

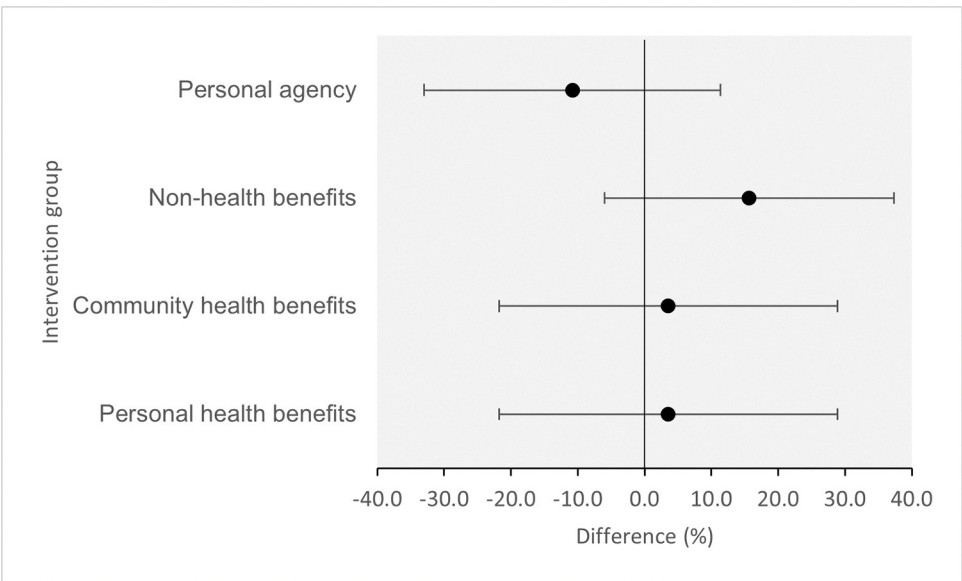

**Fig 3. Difference in intention to vaccinate in each intervention group compared to the control group in hesitant participants (sub-analysis).**

Table 5 shows COVID-19 vaccine booster dose beliefs, compared between groups. Three of the intervention groups showed differences ≥5 percentage points in their agreement with some beliefs compared to the control, although there is a lack of evidence in support of these findings. Participants who viewed the message emphasising personal health benefits showed a higher agreement compared to the control group with beliefs about COVID-19 booster dose safety (percentage point diff: 5.1, 95% CI -7.6, 17.9, $p = 0.424$) and the necessity of booster doses for protecting other people's health (percentage point diff: 7.0, 95% CI -5.4, 19.3, $p = 0.266$). Participants who viewed the message emphasising non-health benefits showed a higher agreement compared to the control with beliefs about booster doses being effective (percentage point diff: 7.6, 95% CI -4.4, 19.6, $p = 0.210$) and the necessity of booster doses for protecting other people's health (percentage point diff: 6.7, 95% CI -5.2, 18.6, $p = 0.266$). Participants who viewed the message emphasising personal agency showed lower agreement compared to the control with beliefs about COVID-19 booster dose safety (percentage point diff: -5.2, 95% CI -17.7, 7.2, $p = 0.404$).

In the sub-analysis of hesitant participants, fewer hesitant participants agreed with beliefs compared to the whole study population. A slight majority of hesitant participants agreed that booster doses protect their health (55.7%, 93/167) and the health of others (53.9%, 90/167). In contrast, a slight minority of hesitant participants agreed that booster doses are safe (43.1%, 72/167), that booster doses prevent disease (48.5%, 81/167), or that they were at risk of getting COVID-19 without a booster dose (46.1%, 77/167). Looking at specific intervention groups, there were more differences ≥5 percentage points compared to the control in hesitant participants compared to the whole study population, although there is a lack of evidence in support of these findings. The full results are included in S1 File.

## Discussion

This experiment found some evidence that messages emphasising non-health benefits of getting a COVID-19 booster dose, like travelling, enjoying family occasions like weddings, and

**Table 5. Comparing beliefs between intervention groups and the control group.**

| Belief | Strongly/slightly agree | | Diff§ | 95% lower | 95% upper | p |
|---|---|---|---|---|---|---|
| | n | (%) | | | | |
| **"Booster doses of COVID-19 vaccine are safe"** | | | | | | |
| Control (n = 109) | 81 | (74.3%) | ref | | | |
| Personal health benefits (n = 73) | 58 | (79.5%) | 5.1* | -7.6 | 17.9 | 0.424 |
| Community health benefits (n = 80) | 80 | (75.0%) | 0.7 | -12.0 | 13.4 | 0.914 |
| Non-health benefits (n = 83) | 61 | (73.5%) | -0.8 | -13.5 | 11.9 | 0.898 |
| Personal agency (n = 97) | 37 | (69.1%) | -5.2* | -17.7 | 7.2 | 0.404 |
| All (n = 442) | 317 | (71.7%) | | | | |
| **"Booster doses of COVID-19 vaccine do a good job preventing disease"** | | | | | | |
| Control (n = 109) | 81 | (74.3%) | ref | | | |
| Personal health benefits (n = 73) | 55 | (75.3%) | 1.0 | -12.0 | 14.1 | 0.875 |
| Community health benefits (n = 80) | 58 | (72.5%) | -0.2 | -14.7 | 11.1 | 0.780 |
| Non-health benefits (n = 83) | 68 | (81.9%) | 7.6* | -4.4 | 19.6 | 0.210 |
| Personal agency (n = 97) | 71 | (73.2%) | -1.1 | -13.3 | 11.0 | 0.856 |
| All (n = 442) | 333 | (75.3%) | | | | |
| **"Booster doses of COVID-19 vaccine are necessary to protect my health"** | | | | | | |
| Control (n = 109) | 86 | (78.9%) | ref | | | |
| Personal health benefits (n = 73) | 61 | (83.6%) | 4.7 | -7.1 | 16.5 | 0.434 |
| Community health benefits (n = 80) | 60 | (75.0%) | -3.9 | -16.1 | 8.3 | 0.528 |
| Non-health benefits (n = 83) | 67 | (80.7%) | 1.8 | -9.8 | 13.4 | 0.756 |
| Personal agency (n = 97) | 74 | (76.3%) | -2.6 | -14.1 | 8.9 | 0.653 |
| All (n = 442) | 348 | (78.7%) | | | | |
| **"Booster doses of COVID-19 vaccine are necessary to protect other people's health"** | | | | | | |
| Control (n = 109) | 82 | (75.2%) | ref | | | |
| Personal health benefits (n = 73) | 60 | (82.2%) | 7.0* | -5.4 | 19.3 | 0.266 |
| Community health benefits (n = 80) | 60 | (75.0%) | -0.2 | -12.9 | 12.4 | 0.971 |
| Non-health benefits (n = 83) | 68 | (81.9%) | 6.7* | -5.2 | 18.6 | 0.266 |
| Personal agency (n = 97) | 72 | (74.2%) | -1.0 | -13.0 | 11.0 | 0.869 |
| All (n = 442) | 342 | (77.4%) | | | | |
| **"If I don't get a booster dose of COVID-19 vaccine, I may get COVID-19"** | | | | | | |
| Control (n = 109) | 67 | (61.5%) | ref | | | |
| Personal health benefits (n = 73) | 47 | (64.4%) | 2.9 | -11.6 | 17.4 | 0.690 |
| Community health benefits (n = 80) | 52 | (65.0%) | 3.5 | -10.6 | 17.6 | 0.619 |
| Non-health benefits (n = 83) | 54 | (65.1%) | 3.6 | -10.3 | 17.5 | 0.609 |
| Personal agency (n = 97) | 62 | (63.9%) | 2.5 | -10.9 | 15.8 | 0.717 |
| All (n = 442) | 282 | (63.8%) | | | | |

§ Percentage point difference

*Indicates a difference ≥5 percentage points between the intervention group and the control group

seeing family and friends, may increase intention to vaccinate, especially in hesitant populations. In this study, intention was qualitatively higher in participants who viewed messages about non-health benefits compared to those who viewed messages about health benefits for themselves and the community. This is somewhat consistent with findings from an experiment in US adults [25], which found some evidence of persuasive messages about non-health benefits being effective, however not more so than other persuasive strategies. This finding from the current study could be explained by the point in time in which this experiment was

conducted. At the time, the Australian population had endured almost two years of public health restrictions, including lockdowns and international and domestic border closures. Being able to live more normally, free from restrictions, may have been top of mind, and hence what participants found the most persuasive. Equally, participants may have found the benefits of vaccination beyond the prevention of severe disease more appealing, especially younger participants. While evidence from a cross-national study early in the pandemic suggested age was not associated with risk perceptions relating to COVID-19 [42], Australian participants in this current study may have experienced or observed others experiencing mild COVID-19 during the first Omicron wave in Australia in December 2021 and thus may not have felt vulnerable to the severe effects of COVID-19.

This study also found some evidence that messages emphasising benefits of COVID-19 vaccine booster doses to people's personal health and the health of the community may increase intention to vaccinate. These findings are consistent with a systematic review of COVID-19 vaccine interventions [22]. Despite qualitatively increasing intention, however, reminding participants about the personal or community health benefits of booster doses was only slightly more effective than informing participants about their eligibility to receive a booster dose (the control message). This is consistent with findings from an experimental study that found that information about where to get a vaccine was not rendered more effective when combined with information about the community or personal benefits of getting vaccinated [43]. In keeping with conclusions from the systematic review [22], reminders about the health benefits of COVID-19 vaccination for one's self and others may have had less of an effect due to participants' over-familiarity with this information after more than 12 months of communication from health authorities.

This study found that messages emphasising personal agency may have negative impacts, especially with hesitant individuals. This finding contrasts with previous findings, which suggest that individual liberty and a sense of personal agency when choosing vaccination may motivate people to vaccinate [18, 19]. The emphasis this message placed on the act of choosing a booster dose as 'the right decision' may help explain this result. Hesitant individuals may have reacted negatively (psychological reactance) [44] to suggestions about how they ought to behave, and what others approve or disapprove of regarding vaccination decisions. Research suggests infringement on freedom is associated with less positive attitudes towards vaccination [45] and, in cases where vaccination mandates are applied, this may lead to lower intentions to vaccinate [46], although findings in this area are mixed [47]. Equally, liberty as a moral value driving vaccination decision-making may vary by country and political views and may not resonate strongly with Australian audiences.

Finally, in terms of COVID-19 vaccine booster dose beliefs, this study found more increased changes in beliefs in hesitant participants compared to the whole study population. This suggests that hesitant participants have beliefs that are possibly less fixed and more amenable to change, a result that has been found in studies about childhood vaccination [48]. This would not necessarily hold for staunch vaccine refusers, but rather individuals who are more undecided or "fence sitters" [49].

These have practical implications for communications from health authorities and other stakeholders to encourage uptake of COVID-19 booster doses, especially communications directed at individuals consuming vaccination information online. Health authorities should consider using a range of messages; an emphasis on non-health benefits may be particularly useful, although personal and community health benefits may also be effective. Any message should be pre-tested with target audiences; message effectiveness will depend on current circumstances and sentiment. Interventions specifically designed for and targeting hesitant, undecided populations are likely to bear the most fruit. For this population, messages

emphasising personal agency should be used with caution. Until more evidence emerges, caution should also be taken with messages that leverage prescriptive norms that tell people how they ought to act (i.e., what the 'right' vaccination behaviour is). In all instances, messages should be tailored for specific communities, taking into account differences in factors affecting motivation to vaccinate and health literacy. Caution should be used when extrapolating this evidence for use with messages directed at individuals hesitant about other vaccines. Research with the community during the initial rollout of COVID-19 vaccines indicates the vaccines are perceived to be considerably different from other vaccines give their use of novel technologies (such as mRNA) and rapid development. Some individuals who report being accepting of most vaccines have indicated hesitancy in relation to COVID-19 vaccines [29, 30].

Future research could use qualitative methodologies to explore message development and testing to better understand the negative effects observed in this study. Research to better understand the effect of messages that leverage prescriptive norms would be useful. Exploring the effects of messaging on different age groups, as well as messaging that caters for varying literacy levels, would provide evidence to support development of more nuanced and targeted communications. Likewise, developing and testing different message formats beyond written information, such as videos and infographics, and channels beyond a static webpage would also be useful. Research focusing specifically on at-risk groups and hesitant populations would ensure greater vaccine equity and would provide useful insights into intervention effectiveness in key target populations. At-risk groups could include the elderly, pregnant women and people, people from different cultural or Indigenous communities, and people with disabilities. Several years into the COVID-19 pandemic, it would also be useful to understand 'vaccine information fatigue' (i.e., the extent to which communities have switched off from and appear immune to communication about COVID-19 vaccines), and how to continue encouraging uptake. This is particularly important if health authorities are to consider additional booster doses or an annual COVID-19 vaccine. Furthermore, given the overall high intentions found among the sample in this study, future research may need to shift the focus to interventions that translate intention into vaccination behaviour (i.e., people getting vaccinated). This could involve the use of behavioural nudges, such as text message reminders, which has shown some positive results early in the COVID-19 vaccination rollout, as well as with other vaccines such as influenza [38, 43].

This study has limitations. In this study, participants only saw the message once from a single source; multiple exposures from multiple sources over time may be required for long term changes to intention and behaviour to occur. This study did not measure changes in behaviour, but rather measured vaccination intention as an outcome. While in keeping with similar studies, behaviour in the form of vaccination uptake would provide a more accurate measure of the effectiveness of such interventions. Some of the intervention texts had a slight overlap. There was a clear emphasis on a single area of focus in each text via a bolded title and repetition of the particular benefit throughout the text. Future research, however, should use a manipulation check to verify that participants can identify the condition they are in, as well as checks to assess comprehension. The sample was highly educated, likely a result of the nature of the panel used. Online research panels in general are likely to represent people who are digitally literate and are willing and able to spend time responding to email invitations and surveys [50]. While this group is the target population for this study, these results may not be generalisable to other populations, such as individuals with low health literacy or from culturally and linguistically diverse backgrounds. While the study is underpowered due to a post hoc variation to the protocol, the findings reported as difference in proportion is practically more meaningful and easier to communicate. Participants were quasi-randomised due to use of the sequence to prioritise the least filled group. Given equal distribution of participants among

intervention groups, however, the threat to internal validity is limited. An appropriate, validated measure of hesitancy towards COVID-19 vaccines that did not include intention items was not available at the time of this study. As such, the measure used may not have accurately captured this construct. The study measured group differences rather than using a pre-post within-subject design, and thus did not capture changes in individual participants' intentions after viewing persuasive messages. Checks to assess comprehension would further strengthen the study.

## Conclusions

Health authorities should consider emphasising the broader benefits of vaccination beyond prevention of severe disease to encourage uptake of COVID-19 vaccine booster doses, as well as benefits to people's personal health and the health of the community. Communication using personal agency should be used with caution. The findings of this research can inform future communication about booster doses of COVID-19 vaccines.

## Supporting information

**S1 Checklist. CONSORT 2010 checklist of information to include when reporting a randomised trial\*.**
(DOC)

**S1 File. Supporting information.**
(DOCX)

**S2 File. Protocol for persuasive COVID-19 vaccination message testing.**
(DOCX)

## Author Contributions

**Conceptualization:** Maryke S. Steffens, Bianca Bullivant, Jessica Kaufman, Catherine King, Margie Danchin, Monsurul Hoq, Mathew D. Marques.

**Formal analysis:** Maryke S. Steffens, Monsurul Hoq, Mathew D. Marques.

**Funding acquisition:** Maryke S. Steffens.

**Investigation:** Maryke S. Steffens, Jessica Kaufman, Mathew D. Marques.

**Methodology:** Maryke S. Steffens, Bianca Bullivant, Jessica Kaufman, Catherine King, Margie Danchin, Monsurul Hoq, Mathew D. Marques.

**Project administration:** Bianca Bullivant.

**Visualization:** Maryke S. Steffens.

**Writing – original draft:** Maryke S. Steffens.

**Writing – review & editing:** Maryke S. Steffens, Bianca Bullivant, Jessica Kaufman, Catherine King, Margie Danchin, Monsurul Hoq, Mathew D. Marques.

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
