## [Decision Letter · Decision Letter 0]

8 Jan 2023

PONE-D-22-28700Testing persuasive messages about booster doses of COVID-19 vaccines on intention to vaccinate in Australian adults: A randomised controlled trialPLOS ONE

Dear Dr. Steffens,

Thank you for submitting your manuscript to PLOS ONE. After careful consideration, we feel that it has merit but does not fully meet PLOS ONE’s publication criteria as it currently stands. Therefore, we invite you to submit a revised version of the manuscript that addresses the points raised during the review process.

The reviewers have offered a number of points of criticism which will need to be addressed if you choose to resubmit to PLOS ONE. There were several concerns about the methodology, design, and statistics which were considered major criticisms necessitating a major revision. There were also a number of more minor comments and critiques from the reviewers.

We look forward to receiving your revised manuscript.

Kind regards,

Stephen R. Walsh, MDCM

Academic Editor

PLOS ONE

2. CT registration (ACTRN12622001404718, https://www.anzctr.org.au/ACTRN12622001404718.aspx) and confirmation that all related CTs are registered, using send back in ITC desk notes. At RTC, please check the authors' response and ping me (aschaefer@plos.org) if the authors do not address this.

Reviewers' comments:

Reviewer's Responses to Questions

**Comments to the Author**

1. Is the manuscript technically sound, and do the data support the conclusions?

Reviewer #1: Yes

Reviewer #2: No

Reviewer #3: Yes

Reviewer #4: Partly

2. Has the statistical analysis been performed appropriately and rigorously? 

Reviewer #1: Yes

Reviewer #2: No

Reviewer #3: Yes

Reviewer #4: Yes

3. Have the authors made all data underlying the findings in their manuscript fully available?

Reviewer #1: Yes

Reviewer #2: Yes

Reviewer #3: Yes

Reviewer #4: Yes

4. Is the manuscript presented in an intelligible fashion and written in standard English?

Reviewer #1: Yes

Reviewer #2: Yes

Reviewer #3: Yes

Reviewer #4: Yes

5. Review Comments to the Author

Please use the space provided to explain your answers to the questions above. You may also include additional comments for the author, including concerns about dual publication, research ethics, or publication ethics. 

Reviewer #1: This is a really well written and interesting manuscript. I have minor suggested revisions for consideration:

1. When describing participants, please provide more information on the "eligible members" who received an email invitation to participate. Is this a random sample? Convenience sample? Your sample was fairly highly educated, is this due to the sampling strategy utilized? Is this a representative sample?

2. Regarding the intervention, what was your rationale for choosing written only? In the discussion, you mention video and other messaging channels. What was your rationale for not considering those here?

3. Considerations for the Discussion section:

a. There is no discussion of how your design may have impacted responses. Did you take into account a recency effect bias? If not, this should be listed as a potential limitation.

b. Also, we know that individuals often need to see messages multiple times from multiple sources for behavioral change to occur. How does this impact your findings? This should be in the discussion.

c. From the HIV literature, we know that willingness/intent does not necessarily translate to action. This is important and should be discussed as well.

4. The sentence on lines 268 and 269 is a repeated sentence.

5. Thank you for pointing out that messages should be pre-tested on target audiences. So critical!

Reviewer #2: Include the power calculation in the text. However, the sample size calculation is not correct. ANOVA is not suitable for dichotomous outcome and testing two proportions. This poor design poses a question about the conclusion as there is no way to know whether the non-significance is due to lack of statistical power.

In addition, the Cohen’s criteria for small or effect size is empirical and better not be used as guidance for sample size unless there is clinical evidence to support it.

In the protocol, ANOVA was mentioned to analyze the primary outcome which is not correct. However, in the text, “a test of two proportions” is not clear. For 5 groups, a test of two proportions only is not a good approach since it does not use all data in one model.

The sample sizes after randomization were severely unbalanced, ranging from 84-115, a 30% difference. This raises another major concern about the execution of the trial.

The flow chart needs to be modified. Check the journal requirement. Separate the frequencies of failed the attention check and did not complete.

Reviewer #3: Overall, I found the article to be complete and did provide some new information that may be helpful for those medical professionals dealing with vaccine hesitancy. I recommend publishing with the following minor edits addressed.

1. Please consider adding information regarding the Research company's accredited online panel. Is this a true representation of the population?

2. Please consider additional information regarding the intervention and control messages. Were these messages created or adapted by the research team? At what literacy level were the messages written? How often did the survey participants see the messages- just one time for 30 seconds ?

3. Is there any attempt to determine how many of the participants actually received the booster, further validating the "intention to vaccinate".

4. Please consider more discussion of how the intervention messages could be used on a wider basis for vaccine hesitancy.

Reviewer #4: This paper uses a survey experiment to examine which communication interventions might help to improve the willingness of receiving a booster vaccination against COVID-19. I enjoyed reading it and find it to be timely and relevant. However, I have some methodological and theoretical concerns and cannot recommend the paper for publication in its current form.

Abstract

The authors state “In addition to emphasizing severe disease prevention, health authorities should highlight broader non-health benefits to encourage COVID-19 vaccine booster uptake”. However, the authors do not specifically test whether the intervention messages have any impact in addition to messages emphasizing severe disease prevention. It could also be the case that these intervention messages do not help much if broad information about the severity of COVID-19 is also provided alongside them.

Introduction

It would be insightful to have a brief discussion on the different types of messages that could or should be provided to increase vaccination uptake. So far, the four chosen message types are not motivated theoretically or methodologically at all which is especially problematic since the messages overlap in parts (e.g., community health benefits also includes an argument about personal health benefits (“getting a booster vaccination not only protects you”) and personal agency also includes an argument of community health benefits (“and lets you protect the people you care about”)).

Since the non-health benefit message addresses potential restrictions that might eventually return, it would be helpful if the authors could provide more information about the exact situation in Australia at the time of data collection. Which restrictions were still active which were not? Which were the restrictions the respondents thought of when hearing “you’re helping to reduce the chance that restrictions return”? How likely was is that these restrictions would actually be re-established?

Methods

Information about the sample is rather vague. Was this a nationally representative sample? Who was eligible to participate? Do the authors believe that the results also hold for the whole adult population in Australia? The authors should discuss this in more detail and provide more information about the sample as well as about unit non-response.

The four messages about personal health benefits, community health benefits, non-health benefits and personal agency are never conceptualized or theorized in the manuscript. It is also not shown or discussed that the messages actually address these four different topics or that respondents understand the messages correctly. The authors should provide more information on how the messages were validated.

The authors should provide more detailed descriptive information about the post-intervention measures. This is especially important because the primary outcome variable is dichotomized. The authors should also discuss in more detail how the dichotomizing of the variable impacts the interpretation of the results.

The authors should be more precise whether they are talking about percent or percentage points. I also find it very odd to argue that differences >5% could be understood as substantial regardless of the p-value. There is no statistical justification for that. Also, the authors later highlight differences >5 percentage points in theirs tables which makes even less sense.

Results

Overall, the small sample size is a problem since it allows only very weak results and limits the possibilities for further analysis a lot.

For example, it is well known that specific person groups respond differently to specific messages (e.g., with respect to age or education) and the authors also include this argumentation themselves (see line 363 on p.19). However, in the analysis this is not reflected in any way.

The authors focus on hesitant respondents and argue that the messages might have a different effect for this group. Since the distribution of hesitancy differs for the different message-types (32.9 to 43.3), the authors should consider also reporting the results for the non-hesitant respondents. However, for the sub-group analysis, the sample is underpowered but the results are reported as though they were not.

There is a typo in Table 5 (-0,18).

Discussion

The authors state that there is a discrepancy between the findings in this study and findings from an US experiment that non-health benefits messages were not more effective than other persuasive strategies. However, I do not see this discrepancy. In this paper there are also no substantial differences between the different persuasive strategies (with the exception of personal agency).

I appreciate the authors discussing the possibility of the personal agency message being corrupted due to the “right decision” part. However, later in the paragraph and in the Conclusion the authors treat this speculation as if it was a finding. It is still completely unclear how respondents would react if the “right decision” part would be missing.

As mentioned above, it would be very helpful to find a paragraph on how the messages were constructed and to also see the results for the “accepting” group as well.

Conclusion

The results do not back up the statement “These messages were more effective with vaccine hesitant individuals.”.

6. PLOS authors have the option to publish the peer review history of their article (what does this mean?). If published, this will include your full peer review and any attached files.

Reviewer #1: **Yes: **Michele Peake Andrasik

Reviewer #2: No

Reviewer #3: No

Reviewer #4: No

---

## [Author Response · Author response to Decision Letter 0]

9 Mar 2023

Please refer to the attached 'response to reviewers' document. I have also pasted the unformatted version of this text below.

Response to reviewers for ‘Testing persuasive messages about booster doses of COVID-19 vaccines on intention to vaccinate in Australian adults: A randomised controlled trial’

Reviewer #1

1.1 This is a really well written and interesting manuscript. I have minor suggested revisions for consideration.

Our response: Thank you for your encouraging assessment of our paper.

1.2 When describing participants, please provide more information on the "eligible members" who received an email invitation to participate. Is this a random sample? Convenience sample? Your sample was fairly highly educated, is this due to the sampling strategy utilized? Is this a representative sample?

Our response: The online panel company (QOR) sent email invitations to a random sample of its members. We have updated the manuscript to reflect this.

The text in the Methods now reads:

“Research company Quality Online Research (QOR) recruited a random sample of participants via email invitation from its accredited online panel.”

We agree that our sample was fairly highly educated. This likely reflects the nature of the panel; previous research has shown that online panels are more likely to represent people who are digitally literate and are willing and able to spend time responding to email invitations and surveys. Our target population for the intervention was digitally literate individuals likely to read emails and other digital written communication from health authorities. We intended our sample to be representative of this target population. Our sample was not nationally representative, as that was not our intention. We acknowledge that this means our findings may not be generalisable to other populations such as those with lower health literacy. We have updated the manuscript to note this as a limitation of the study.

The text in the Discussion now reads:

“The sample was highly educated, likely a result of the nature of the panel used. Online research panels in general are likely to represent people who are digitally literate and are willing and able to spend time responding to email invitations and surveys [ref]. While this group is the target population for this study, these results may not be generalisable to other populations, such as individuals with low health literacy or from culturally and linguistically diverse backgrounds.”

1.3 Regarding the intervention, what was your rationale for choosing written only? In the discussion, you mention video and other messaging channels. What was your rationale for not considering those here?

Our response: Our rationale for choosing to test a written intervention (versus video or other types of media) was to inform written communications (e.g. emails) from health authorities and other stakeholders. These were used regularly to communicate with the public in Australia during the COVID-19 vaccine rollout. Testing written interventions also enabled us to deliver timely results to stakeholders to inform their communication approaches during the COVID-19 vaccination rollout. We agree with the reviewer that it is important to also test video and other types of media and have suggested this as a future area of research in the Discussion. We have also modified the text in the Introduction to give more explanation of why we tested written interventions.

The text the Introduction now reads:

“Such evidence can inform communications from health authorities and other stakeholders to encourage uptake of COVID-19 booster doses, especially written communications (e.g. emails), which were used to communicate with the public in Australia during the COVID-19 vaccine rollout.”

1.4 Considerations for the Discussion section: There is no discussion of how your design may have impacted responses. Did you take into account a recency effect bias? If not, this should be listed as a potential limitation.

Our response: From Reviewer 1’s comment, we are unclear how they feel a recency effect bias may relate to our study. Our interpretation is that Reviewer 1 is asking us to reflect how intention to vaccinate may change over time after an intervention, and whether our study design takes this into consideration. We acknowledge that there are questions around how long the effect of an intervention may last. We have updated the manuscript to include a limitation noting that our participants only saw the intervention once, and that intention to vaccinate may change over time. Please also see our response in 1.5.

The text in the Discussion now reads:

“In this study, participants only saw the message once from a single source; multiple exposures from multiple sources over time may be required for long term changes to intention and behaviour to occur.”

1.5 Also, we know that individuals often need to see messages multiple times from multiple sources for behavioral change to occur. How does this impact your findings? This should be in the discussion.

Our response: In this study, we did not set out to measure behavioural change, but rather looked at intention to vaccinate. Various studies have shown that a single exposure to a message can affect intention to vaccinate, although there are questions around how long the effect may last. We agree that for a lasting effect, including on behaviour (and shifting this is the ultimate aim), exposing individuals to multiple messages from multiple sources will likely have the best outcome. We have addressed this as a limitation of the study. Please see 1.4 for changes made to the manuscript.

1.6 From the HIV literature, we know that willingness/intent does not necessarily translate to action. This is important and should be discussed as well.

Our response: We wholeheartedly agree that it would be more informative to measure change in behaviour in the form of vaccination uptake, rather than measuring intention to vaccinate. This is, however, a difficult outcome to measure in such studies. We acknowledge that this is an inadequacy of the study design and have listed this as a limitation. 

The text in the Discussion now reads:

“This study did not measure changes in behaviour, but rather measured vaccination intention as an outcome. While in keeping with similar studies, behaviour in the form of vaccination uptake would provide a more accurate measure of the effectiveness of such interventions.”

1.7 The sentence on lines 268 and 269 is a repeated sentence.

Our response: Thank you, we have removed this repetition.

The text in the Methods now reads:

“Most participants (84.6%, 374/442) indicated a positive intention to get a booster dose of COVID-19 vaccine. Table 3 shows intention to vaccinate, compared between groups. All intervention groups had…”

1.8 Thank you for pointing out that messages should be pre-tested on target audiences. So critical!

Our response: Thank you, we agree. In our experience, messages sometimes need to be changed considerably.

No changes were made to the text. 

Reviewer #2

2.1 Include the power calculation in the text.

Our response: We have now included the power calculation in the main manuscript and have removed this from the supporting materials.

The text in Methods now reads:

“The study aimed to recruit 480 participants to ensure a sample size of 430 participants, allowing for a drop-out/poor quality response rate of approximately 10%. The sample size was calculated using G*Power (an a priori power analysis tool) to estimate an effect size of Cohen’s d = 0.2 (the difference between two independent means) [ref] with a specified power of 80% at 0.01 level of significance using analysis of variance (ANOVA) tests to compare outcomes between intervention groups, as specified in the pre-registered protocol. In line with the post hoc variation to analyse participant’s intention to vaccinate and belief items as dichotomous variables, the sample size of 85 participants per intervention group was adequate to estimate a difference in proportion of 15 percentage points or more between two groups with a confidence level (CI) of 95% and power of 80%.”

2.2 However, the sample size calculation is not correct. ANOVA is not suitable for dichotomous outcome and testing two proportions. This poor design poses a question about the conclusion as there is no way to know whether the non-significance is due to lack of statistical power.

Our response: We originally calculated the sample size in anticipation of using ANOVA tests as specified in the protocol to compare mean intention or belief scores between the groups. However, in response to the skewed distribution of responses, we made a post hoc variation to the analysis plan, analysing the outcome measures as dichotomous variables using tests of two proportions. We have now included a post hoc power calculation and the statistically significant difference we could estimate for this sample size. 

The reviewer rightly pointed out that the statistical significance based on an arbitrary cut-off of 0.5 is dependent on the sample size. Hence we highlighted the difference of 5 percentage points in vaccine intention from a practical significance point and considered the upper and lower limits of the confidence intervals with respect to the difference. This is now noted in the manuscript. Please see 2.1 above for changes made to the text. 

2.3 In addition, the Cohen’s criteria for small or effect size is empirical and better not be used as guidance for sample size unless there is clinical evidence to support it.

Our response: We appreciate the reviewer’s comments regarding Cohen’s d. In line with the protocol variation, we have highlighted differences of ≥5 percentage points from a practical (clinical) significance point based on previous research. We have acknowledged that the sample size is not adequate to estimate a statistically significant difference of 5 percentage points. Please see 2.1 above for changes made to the text.

2.4 In the protocol, ANOVA was mentioned to analyze the primary outcome which is not correct.

Our response: We made a post hoc variation to the analysis plan in response to the skewed distribution of responses. Please refer to the response to 2.2 above.

2.5 However, in the text, “a test of two proportions” is not clear. For 5 groups, a test of two proportions only is not a good approach since it does not use all data in one model.

Our response: We apologise for the grammatical mistake. We have now corrected this to clarify that each of the four intervention groups were compared with the control group using tests of two proportions. Using four separate tests of two proportions, comparing each of the intervention groups with the control, is the same as using a binomial regression to calculate risk difference for each of the four intervention groups with control group as the reference. 

The text in the Methods now reads:

“The difference in proportion of participants intending to receive a COVID-19 vaccine and agreeing with COVID-19 vaccine booster dose belief statements was compared between the control group and each of the four intervention groups using tests of two proportions.” 

2.6 The sample sizes after randomization were severely unbalanced, ranging from 84-115, a 30% difference. This raises another major concern about the execution of the trial.

Our response: We apologise for the misunderstanding, due to a grammatical error. As outlined in 2.5, we did not compare the outcome among five intervention groups, rather we compared the outcome between each of the four interventions with the control group. In this case, the control group with a sample size of 109 would be more than any of the intervention groups, which is common in RCTs. 

We conducted an analysis of participants at baseline, and did not find any differences in age, gender, education, state, completion of second dose, or hesitancy between groups. We have reported this in the Results. Please see 2.5 for changes made to the text. 

2.7 The flow chart needs to be modified. Check the journal requirement. Separate the frequencies of failed the attention check and did not complete.

Our response: The participants listed in the ‘excluded’ section of the flow chart were those who failed the attention check. We have updated the flow chart to indicate this.

Please refer to the updated Figure 1.

Reviewer #3

3.1 Overall, I found the article to be complete and did provide some new information that may be helpful for those medical professionals dealing with vaccine hesitancy. I recommend publishing with the following minor edits addressed. 

Our response: Thank you for your encouraging assessment of our paper.

3.2 Please consider adding information regarding the Research company's accredited online panel. Is this a true representation of the population?

Our response: We have added information to the manuscript about the research company’s accredited panel.

The text in the Methods now reads:

“The QOR panel has >85,000 active members; original panel members were recruited via the Australia Post Lifestyle Survey, distributed to all Australian households. Ongoing recruitment is by invitation only. The panel reflects Australian Bureau of Statistics census data by age, gender, and state.”

As outlined in 1.2, our target population for the intervention was digitally literate individuals likely to read emails and other digital written communication from health authorities. We intended our sample to be representative of this target population. Our sample was not nationally representative, as that was not our intention. We acknowledge that this means our findings may not be generalisable to other populations, such as those with low health literacy. We have updated the manuscript to note this as a limitation of the study.

The text in the Discussion now reads:

“The sample was highly educated, likely a result of the nature of the panel used. Online research panels in general are likely to represent people who are digitally literate and are willing and able to spend time responding to email invitations and surveys [ref]. While this group is the target population for this study, these results may not be generalisable to other populations, such as individuals with low health literacy or from culturally and linguistically diverse backgrounds.”

3.3 Please consider additional information regarding the intervention and control messages. Were these messages created or adapted by the research team? At what literacy level were the messages written? How often did the survey participants see the messages- just one time for 30 seconds?

Our response: The messages were a modified version of a public email communication campaign disseminated by an Australian health authority in late 2021. We have updated the manuscript to indicate that the research team made these modifications based on the current state of evidence regarding the types of messages that may have an effect on intention to vaccinate. 

The text in Methods now reads:

“These modifications were made by the research team, based on the current state of evidence on the types of messages that may have an effect on intention to vaccinate.”

Participants saw the messages just once for 30 seconds. As noted above in 1.5, for a lasting effect, exposing individuals to multiple messages from multiple sources may have better outcomes. We have addressed this as a limitation of the study.

The text in the Discussion now reads:

“In this study the research participants only saw the message once, from a single source; multiple exposures from multiple sources over time may be required for long term changes to intention and behaviour to occur.”

The literacy levels of the messages were not tested. We have listed this as a limitation of the study and added testing messages for varying literacy levels as an area for future research. 

The text in the Discussion now reads:

“Some of the intervention texts had a slight overlap. There was a clear emphasis on a single area of focus in each text via a bolded title and repetition of the particular benefit throughout the text. Future research, however, should use a manipulation check to verify that participants can identify the condition they are in, as well as checks to assess comprehension.”

And 

“Exploring the effects of messaging on different age groups, as well as messaging that caters for varying literacy levels, would provide evidence to support development of more nuanced and targeted communications.”

3.4 Is there any attempt to determine how many of the participants actually received the booster, further validating the "intention to vaccinate".

Our response: As noted in 1.6, we agree that it would be more informative to measure change in behaviour in the form of vaccination uptake, rather than measuring intention to vaccinate. This is, however, a difficult outcome to measure in such studies. We acknowledge that this is an inadequacy of the study design and have listed this as a limitation. 

The text in the Discussion now reads:

“This study did not measure changes in behaviour, but rather measured vaccination intention as an outcome. While in keeping with similar studies, behaviour in the form of vaccination uptake would provide a more accurate measure of the effectiveness of such interventions.”

3.5 Please consider more discussion of how the intervention messages could be used on a wider basis for vaccine hesitancy.

Our response: We have updated the manuscript to indicate that caution should be used when extrapolating evidence from this study to developing interventions for vaccine hesitancy more broadly. Our rationale is that COVID-19 vaccines are perceived quite differently from other vaccines, and so further research should test interventions for specific vaccines and scenarios.

The text in the Discussion now reads:

“Caution should be used when extrapolating this evidence for use with messages directed at individuals hesitant about other vaccines. Research with the community during the initial rollout of COVID-19 vaccines indicates the vaccines are perceived to be considerably different from other vaccines give their use of novel technologies (such as mRNA) and rapid development. Some individuals who report being accepting of most vaccines have indicated hesitancy in relation to COVID-19 vaccines [refs].”

Reviewer #4

4.1 This paper uses a survey experiment to examine which communication interventions might help to improve the willingness of receiving a booster vaccination against COVID-19. I enjoyed reading it and find it to be timely and relevant. However, I have some methodological and theoretical concerns and cannot recommend the paper for publication in its current form. 

Our response: Thank you for taking the time to review our manuscript.

4.2 Abstract: The authors state “In addition to emphasizing severe disease prevention, health authorities should highlight broader non-health benefits to encourage COVID-19 vaccine booster uptake”. However, the authors do not specifically test whether the intervention messages have any impact in addition to messages emphasizing severe disease prevention. It could also be the case that these intervention messages do not help much if broad information about the severity of COVID-19 is also provided alongside them. 

Our response: Thank you for this point, this indicates to us that the language we have used is unclear. We have amended the manuscript to make the Abstract clearer.

The text in the Abstract now reads:

“Health authorities should highlight non-health benefits to encourage COVID-19 vaccine booster uptake but use messages emphasising personal agency with caution.”

4.3 Introduction: It would be insightful to have a brief discussion on the different types of messages that could or should be provided to increase vaccination uptake. 

Our response: In the introduction (4th paragraph), we provide a summary of current evidence about the effectiveness of different types of persuasive messages in increasing intention to vaccinate. This includes messages emphasising personal health benefits, benefits to community health, and non-health benefits. Unfortunately there is limited evidence on whether this translates to changes in behaviour, i.e. uptake of vaccines. We note that this is a limitation of these types of studies and have amended the manuscript to acknowledge this.

The text in the Discussion now reads:

“This study did not measure changes in behaviour, but rather measured vaccination intention as an outcome. While in keeping with similar studies, behaviour in the form of vaccination uptake would provide a more accurate measure of the effectiveness of such interventions.”

4.4 So far, the four chosen message types are not motivated theoretically or methodologically at all which is especially problematic since the messages overlap in parts (e.g., community health benefits also includes an argument about personal health benefits (“getting a booster vaccination not only protects you”) and personal agency also includes an argument of community health benefits (“and lets you protect the people you care about”)). 

Our response: We chose the four message types in response to research evidence available at the time on effective messaging relating to COVID-19 vaccines. Our intention was to address a gap in knowledge regarding the types of messages that have the greatest effect on intention to receive a booster dose of COVID-19 vaccine. We acknowledge that there is a small amount of overlap in some of the intervention texts. There is, however, clear emphasis on a single area of focus in each of the intervention texts via the bolded title and via repetition of the particular benefit throughout the text. We acknowledge that conducting a manipulation check to verify identification of the condition that participants were in would strengthen this study, and have added this as a limitation in the manuscript. 

The text in the Discussion now reads: 

“Some of the intervention texts had a slight overlap. There was a clear emphasis on a single area of focus in each text via a bolded title and repetition of the particular benefit throughout the text. Future research, however, should use a manipulation check to verify that participants can identify the condition they are in, as well as checks to assess comprehension.”

4.5 Since the non-health benefit message addresses potential restrictions that might eventually return, it would be helpful if the authors could provide more information about the exact situation in Australia at the time of data collection. Which restrictions were still active which were not? Which were the restrictions the respondents thought of when hearing “you’re helping to reduce the chance that restrictions return”? How likely was is that these restrictions would actually be re-established? 

Our response: We have added further information to the manuscript about the situation in Australia with regards to COVID-19 restrictions at the time of data collection. We cannot say with any certainty how likely it was that restrictions would be re-established, or what people’s perceptions of that likelihood were at that time. However, we argue that given the pandemic state of emergency was still active at the time, restrictions had been repeatedly ramped up and down over the past 20 or so months, and with the threat of an Omicron wave in December 2021, it is reasonable to assume that people were fearful of restrictions returning.

The text in the Methods now reads:

“For context, at the time of data collection, the Australian population had experienced multiple rounds of restrictions on movement, limits on indoor and outdoor gatherings, closure of restaurants, gyms and non-essential retail, and interstate and international border closures. Approximately two months prior to data collection (in October 2021), people living in the states of New South Wales and Victoria had exited strict and lengthy lockdowns in response to the Delta wave. In December 2021, domestic border closures had started to lift, however strict international border closures, put in place in March 2020 to prevent people from leaving and entering the country, were still in place at the time. In December 2021, Australia was anticipating the occurrence of a further Omicron wave, with cases rising rapidly [refs].”

4.6 Methods: Information about the sample is rather vague. Was this a nationally representative sample? Who was eligible to participate? Do the authors believe that the results also hold for the whole adult population in Australia? The authors should discuss this in more detail and provide more information about the sample as well as about unit non-response. 

Our response: The sample was not nationally representative but was not intended to be. The online panel company contacted a random sample from their eligible online panel (adults who had received 2 doses of a COVID-19 vaccine, but not a booster). The results suggest how an online, digitally literate population (such as that in an online panel) may respond to such messages. We have provided more information about the sample in the manuscript. See also our response to 1.6 above. We recruited participants through an external company, which did not provide information about unit non-response.

The text in the Methods now reads:

“Participants were adults 18 years or older residing in Australia who had received at least one primary dose of a COVID-19 vaccine but had not yet received a booster (third) dose and had access to the internet. Research company Quality Online Research (QOR) recruited a random sample of participants via email invitation from its accredited online panel. The QOR panel has >85,000 active members; original panel members were recruited via the Australia Post Lifestyle Survey, distributed to all Australian households. Ongoing recruitment is by invitation only. The panel reflects Australian Bureau of Statistics census data by age, gender, and state.” 

The text in the Discussion now reads:

“Online research panels in general are likely to represent people who are digitally literate and are willing and able to spend time responding to email invitations and surveys [ref]. While this group is the target population for this study, these results may not be generalisable to other populations, such as individuals with low health literacy or from culturally and linguistically diverse backgrounds.”

4.7 The four messages about personal health benefits, community health benefits, non-health benefits and personal agency are never conceptualized or theorized in the manuscript. It is also not shown or discussed that the messages actually address these four different topics or that respondents understand the messages correctly. The authors should provide more information on how the messages were validated. 

Our response: Please refer to response 4.4. 

4.8 The authors should provide more detailed descriptive information about the post-intervention measures. This is especially important because the primary outcome variable is dichotomized. 

Our response: We have updated the manuscript to be clearer about the post-intervention measures.

The text in Methods now reads:

“The primary outcome measure was intention to receive a COVID-19 vaccine booster dose. This was assessed with a single item (‘How likely is it that you will get a booster dose of COVID-19 vaccine?’, with 5 response options: definitely, probably, I’m not sure, probably not, definitely not), consistent with survey questions used in previous research.”

We have pointed the reader to the supporting information to see the secondary outcome measures (belief statements).

The text in Methods now reads:

“Secondary outcome measures were beliefs about COVID-19 vaccine booster doses. These were assessed by asking participants to indicate their agreement with belief statements about COVID-19 vaccine booster dose safety, effectiveness, necessity for protecting one’s own health, necessity for protecting others’ health, and risks associated with not vaccinating (response options were a 5-point scale: strongly disagree, slightly disagree, neither agree nor disagree, slightly agree, strongly agree. See supporting information for belief statements).”

4.9 The authors should also discuss in more detail how the dichotomizing of the variable impacts the interpretation of the results. 

Our response: The intention to vaccinate and beliefs about COVID-19 vaccines were assessed using a 5-point Likert scale and considered as a continuous variable in line with the previous literature. Considering these variables as a continuous variable is convenient in terms of estimating difference in mean between groups and achieving a higher precision with a smaller sample size. As noted in previous responses, we made a post-hoc variation due to the sample being skewed towards accepting participants. This variation consisted of considering vaccine intention and belief as categorical, dichotomising them into favourable and unfavourable outcomes, and estimating difference in proportion (instead of difference in mean intention or belief scores). The benefit of this approach is that it delivers findings with more practical meaning than estimating the difference in mean intention or belief score. It allows us to interpret the findings in terms of the difference in proportion of people between groups, which is practically meaningful and easy to communicate. We acknowledge that the power of the tests of two proportions is relatively low for the sample size calculated for the sample t-test. We have amended the manuscript to acknowledge the impacts of our post-hoc variation.

The text in the Discussion now reads:

“While the study is underpowered due to a post hoc variation to the protocol, the findings reported as difference in proportion is practically more meaningful and easier to communicate.”

4.10 The authors should be more precise whether they are talking about percent or percentage points. 

Our response: We have updated the manuscript and tables to indicate where we are reporting percent (indicated as ‘%’) and where we are reporting percentage point differences. 

The text has been updated in multiple places; please refer to the revised manuscript and tables.

4.11 I also find it very odd to argue that differences >5% could be understood as substantial regardless of the p-value. There is no statistical justification for that. Also, the authors later highlight differences >5 percentage points in theirs tables which makes even less sense.

Our response: The difference in vaccine intention of more than 5 percentage points was informed by previous literature, which we considered a practically meaningful difference from a public health perspective, not from a statistical perspective. As mentioned previously, we have made a post-hoc variation where vaccine intention and belief were dichotomised and difference in proportion was estimated instead of difference in mean intention or belief scores. The sample size calculated for estimating difference in mean was adequate for estimating a difference in proportion of 15 percentage points or more. We have updated the manuscript to indicate this; please refer to our response to 2.1 above. Hence, instead of interpreting our findings based on statistical significance (on the basis of p-value < 0.05, which is also arbitrary), we highlighted the findings where difference was more than 5 percentage points and interpreted the difference in proportion with respect of the upper and lower limit of the confidence intervals. We have updated the manuscript to reflect this. 

The text in the Methods now reads:

“Previous studies have shown that an effect size of ≥5 percentage points may have practical (clinical) significance [refs]. Hence, when reporting differences in intention and beliefs for each intervention group compared to the control group, differences of ≥5 percentage points were considered practically meaningful from a public health perspective, regardless of the p value [ref]. The difference in proportion was interpreted with respect to the upper and lower limit of the confidence intervals.”

4.12 Results: Overall, the small sample size is a problem since it allows only very weak results and limits the possibilities for further analysis a lot.

Our response: The sample size was calculated based on our original analysis plan of using ANOVA. Due to the sample being skewed towards accepting participants, we made post-hoc changes to the analysis plan, which resulted in the study being under-powered. We have amended the manuscript to acknowledge this limitation. 

The text in the Discussion now reads:

“While the study is underpowered due to a post hoc variation in the protocol, the findings reported as difference in proportion is practically more meaningful and easier to communicate.” 

4.13 For example, it is well known that specific person groups respond differently to specific messages (e.g., with respect to age or education) and the authors also include this argumentation themselves (see line 363 on p.19). However, in the analysis this is not reflected in any way.

Our response: We agree that this is an area for further research. We have updated the manuscript to acknowledge this.

The text in the Discussion now reads:

“Exploring the effects of messaging on different age groups, as well as messaging that caters for varying literacy levels, would provide evidence to support development of more nuanced and targeted communications.”

4.14 The authors focus on hesitant respondents and argue that the messages might have a different effect for this group. Since the distribution of hesitancy differs for the different message-types (32.9 to 43.3), the authors should consider also reporting the results for the non-hesitant respondents.

Our response: In our protocol, we did not specify that we would analyse non-hesitant participants. The reason for this is that hesitant individuals are a more important target audience of interventions, given they are likely less confident in COVID-19 vaccines and are less likely to get vaccinated. However, we acknowledge that these results may be of interest to others and have added them into the supporting information.

Please refer to Table S2 in supporting information. 

4.15 However, for the sub-group analysis, the sample is underpowered but the results are reported as though they were not. 

Our response: We have updated the manuscript to acknowledge that the study is underpowered for sub-analysis. Please see response to 4.12 for changes made to the manuscript. We have also modified the language in various places when describing the results of the sub-analysis to indicate a lack of evidence. 

The text in the Results now reads:

“As with the whole population, all intervention groups had a qualitatively higher intention to vaccinate compared to the control group except one (see Figure 3). Participants who viewed the message emphasising non-health benefits showed a higher intention compared to the control group (percentage point diff: 15.6, 95% CI -6.0, 37.3, p = 0.150), while participants who viewed the personal agency message had a lower intention compared to the control group (percentage point diff: -10.8, 95% CI -33.0, 11.4, p = 0.330), although there is a lack of evidence in support of these findings.” 

4.16 There is a typo in Table 5 (-0,18).

Our response: Thank you, we have corrected the typo.

4.17 Discussion: The authors state that there is a discrepancy between the findings in this study and findings from an US experiment that non-health benefits messages were not more effective than other persuasive strategies. However, I do not see this discrepancy. In this paper there are also no substantial differences between the different persuasive strategies (with the exception of personal agency).

Our response: We have removed the reference to a discrepancy in the manuscript.

The text now reads:

“In this study, intention was qualitatively higher in participants who viewed messages about non-health benefits compared to those who viewed messages about health benefits for themselves and the community. This is somewhat consistent with findings from an experiment in US adults [ref], which found some evidence of persuasive messages about non-health benefits being effective, however not more so than other persuasive strategies. This finding from the current study could be explained by the point in time in which this experiment was conducted. At the time, the Australian population had endured almost two years of public health restrictions…”

4.18 I appreciate the authors discussing the possibility of the personal agency message being corrupted due to the “right decision” part. However, later in the paragraph and in the Conclusion the authors treat this speculation as if it was a finding. It is still completely unclear how respondents would react if the “right decision” part would be missing.

Our response: It was not our intention to treat this speculation as a finding. Our intention was to use measured language and flag to communication practitioners that using this approach requires caution, at least until more research is conducted. We have updated the text in the manuscript to be more conservative and to better reflect this position. We have also added text in the Discussion noting the need for future research in this area. 

The text in the Discussion now reads:

For this population, messages emphasising personal agency should be used with caution. Until more evidence emerges, caution should also be taken with especially those messages that leverage prescriptive norms that tell people how they ought to act (i.e., what the ‘right’ vaccination behaviour is).”

and

“Research to better understand the effect of messages that leverage prescriptive norms would be useful.” 

The text in the Conclusion now reads:

“Communication using personal agency should be used with caution.” 

4.19 As mentioned above, it would be very helpful to find a paragraph on how the messages were constructed and to also see the results for the “accepting” group as well.

Our response: The messages were a modified version of a public email communication campaign disseminated by an Australian health authority in late 2021. We have updated the manuscript to indicate that the research team made these modifications based on the current state of evidence regarding the types of messages that may have an effect on intention to vaccinate. Please also refer to our response to 4.4. 

The text in Methods now reads:

“These modifications were made by the research team, based on the current state of evidence on the types of messages that may have an effect on intention to vaccinate.”

As outlined in our response to 4.14, we have now added the results of the non-hesitant participants to the supporting information. 

4.20 Conclusion: The results do not back up the statement “These messages were more effective with vaccine hesitant individuals.”..

Our response: We have removed this sentence from the Conclusion.

5. Response to Journal requirements

5.1 Please ensure that your manuscript meets PLOS ONE's style requirements, including those for file naming.

Our response: We have updated the manuscript to PLOS ONE’s style requirements.

5.2 Confirm that all related CTs are registered.

Our response: A clinical trial from the same overarching study [1] was retrospectively registered with the Australian New Zealand Clinical Trials Registry (ACTRN12622001554752) and is cited in the manuscript.

References

1. Kaufman J, Steffens MS, Hoq M, King C, Marques MD, Mao K, et al. Effect of persuasive messaging about COVID-19 vaccines for 5- to 11-year-old children on parent intention to vaccinate. J Paediatr Child Health. 2023.

---

## [Decision Letter · Decision Letter 1]

13 Apr 2023

PONE-D-22-28700R1Testing persuasive messages about booster doses of COVID-19 vaccines on intention to vaccinate in Australian adults: A randomised controlled trialPLOS ONE

Dear Dr. Steffens,

Thank you for submitting your manuscript to PLOS ONE. After careful consideration, we feel that it has merit but does not fully meet PLOS ONE’s publication criteria as it currently stands. Therefore, we invite you to submit a revised version of the manuscript that addresses the points raised during the review process.

We look forward to receiving your revised manuscript.

Kind regards,

Stephen R. Walsh, MDCM

Academic Editor

PLOS ONE

Journal Requirements:

Reviewers' comments:

Reviewer's Responses to Questions

**Comments to the Author**

1. If the authors have adequately addressed your comments raised in a previous round of review and you feel that this manuscript is now acceptable for publication, you may indicate that here to bypass the “Comments to the Author” section, enter your conflict of interest statement in the “Confidential to Editor” section, and submit your "Accept" recommendation.

Reviewer #1: All comments have been addressed

Reviewer #2: (No Response)

Reviewer #4: (No Response)

2. Is the manuscript technically sound, and do the data support the conclusions?

Reviewer #1: Yes

Reviewer #2: Yes

Reviewer #4: Yes

3. Has the statistical analysis been performed appropriately and rigorously? 

Reviewer #1: Yes

Reviewer #2: N/A

Reviewer #4: Yes

4. Have the authors made all data underlying the findings in their manuscript fully available?

Reviewer #1: Yes

Reviewer #2: Yes

Reviewer #4: Yes

5. Is the manuscript presented in an intelligible fashion and written in standard English?

Reviewer #1: Yes

Reviewer #2: Yes

Reviewer #4: Yes

6. Review Comments to the Author

Reviewer #1: (No Response)

Reviewer #2: 1. Please clarify whether the test of two proportion (sample size and Line 275) is using Chi-square test or a logistic regression. If logistic regression, add details of covariates.

2. Line 254, sample size is based on a test (Chi-square test?), not 95% CI.

3. Figure 1 the format is not correct. Check the examples on PLOSONE.

Reviewer #4: Even though I understand that dichotomizing the outcome variable makes interpretation easier, the authors should provide more information about it. The cut-off point is rather arbitrary – especially for the middle category. To get a better understanding about the effects of the treatments, it would be helpful to see the differences between the treatment groups for the distribution of the original variable as well. It would also help to understand how the treatment affects the intention of the respondents (e.g., the “definitely not” group might not differ at all between groups, but the “I’m not sure” might shift towards “probably”). This is highly relevant since “definitely not” and “I’m not sure” are very different answers which are treated the same by dichotomizing the variable. The authors should at least provide information about the original distribution in the Supplementary Materials.

7. PLOS authors have the option to publish the peer review history of their article (what does this mean?). If published, this will include your full peer review and any attached files.

Reviewer #1: No

Reviewer #2: No

Reviewer #4: No

---

## [Author Response · Author response to Decision Letter 1]

11 May 2023

Reviewer #2

2.1 Please clarify whether the test of two proportion (sample size and Line 275) is using Chi-square test or a logistic regression. If logistic regression, add details of covariates.

Our response: In this study we used the z-test for two proportions since the sample sizes were large enough (> 30 in each group) and assumed that the data follows a normal distribution. We have updated the manuscript to reflect this.

The text in the Methods now reads:

“The difference in proportion of participants intending to receive a COVID-19 vaccine and agreeing with COVID-19 vaccine booster dose belief statements was compared between the control group and each of the four intervention groups using tests of two proportions (z test).”

2.2 Line 254, sample size is based on a test (Chi-square test?), not 95% CI.

Our response: We have updated the manuscript to remove reference to 95% CI and indicate that we used z tests.

The text in the Methods now reads:

“In line with the post hoc variation to analyse participant’s intention to vaccinate and belief items as dichotomous variables, the sample size of 85 participants per intervention group was adequate to estimate a difference in proportion of 15 percentage points or more between two groups using two-sample test of proportion (z-test) with a level of significance of 0.05 and power of 80%.” 

2.3 Figure 1 the format is not correct. Check the examples on PLOSONE.

Our response: We have updated Figure 1 to be in keeping with Schulz KF, Altman DG, Moher D, for the CONSORT Group (2010). CONSORT 2010 Statement: Updated Guidelines for Reporting Parallel Group Randomised Trials. PLoS Med 7(3): e1000251. We have also updated in-text citations of the figures to be in keeping with PLoS ONE requirements.

Reviewer #4

4.1 Even though I understand that dichotomizing the outcome variable makes interpretation easier, the authors should provide more information about it. The cut-off point is rather arbitrary – especially for the middle category. To get a better understanding about the effects of the treatments, it would be helpful to see the differences between the treatment groups for the distribution of the original variable as well. It would also help to understand how the treatment affects the intention of the respondents (e.g., the “definitely not” group might not differ at all between groups, but the “I’m not sure” might shift towards “probably”). This is highly relevant since “definitely not” and “I’m not sure” are very different answers which are treated the same by dichotomizing the variable. The authors should at least provide information about the original distribution in the Supplementary Materials.

Our response: We have provided information about the original distribution of the intention variable by intervention group in the supporting information. 

The text in the Methods now reads:

“In response to the skewed distribution of responses, a post hoc variation to the pre-registration was made to analyse intention to vaccinate and participant responses to belief items as dichotomous variables. See supporting information for original frequency distribution of intention to vaccinate.”

See the supporting information for the new table showing the original frequency distribution of intention to vaccinate. 

Journal Requirements

1.1 Please review your reference list to ensure that it is complete and correct. If you have cited papers that have been retracted, please include the rationale for doing so in the manuscript text, or remove these references and replace them with relevant current references. Any changes to the reference list should be mentioned in the rebuttal letter that accompanies your revised manuscript. If you need to cite a retracted article, indicate the article’s retracted status in the References list and also include a citation and full reference for the retraction notice.

Our response: We have reviewed our reference list and can confirm that it is complete and correct.

---

## [Editor Report · Decision Letter 2]

24 May 2023

Testing persuasive messages about booster doses of COVID-19 vaccines on intention to vaccinate in Australian adults: A randomised controlled trial

PONE-D-22-28700R2

Dear Dr. Steffens,

We’re pleased to inform you that your manuscript has been judged scientifically suitable for publication and will be formally accepted for publication once it meets all outstanding technical requirements.

Kind regards,

Stephen R. Walsh, MDCM

Academic Editor

PLOS ONE

---

## [Editor Report · Acceptance letter]

26 May 2023

PONE-D-22-28700R2 

Testing persuasive messages about booster doses of COVID-19 vaccines on intention to vaccinate in Australian adults: A randomised controlled trial 

Dear Dr. Steffens:

I'm pleased to inform you that your manuscript has been deemed suitable for publication in PLOS ONE. Congratulations! Your manuscript is now with our production department. 

Kind regards, 

on behalf of

Dr. Stephen R. Walsh 

Academic Editor

PLOS ONE